# Automated structure refinement of macromolecular assemblies from cryo-EM maps using Rosetta

Ray Yu-Ruei Wang[1,2†], Yifan Song[2‡], Benjamin A Barad[3,4], Yifan Cheng[5,6], James S Fraser[3], Frank DiMaio[2,7*]

[1]Graduate Program in Biological Physics, Structure and Design, University of Washington, Seattle, United States; [2]Department of Biochemistry, University of Washington, Seattle, United States; [3]Department of Bioengineering and Therapeutic Science, University of California, San Francisco, San Francisco, United States; [4]Graduate Group in Biophysics, University of California, San Francisco, San Francisco, United States; [5]Keck Advanced Microscopy Laboratory, University of California, San Francisco, San Francisco, United States; [6]Department of Biochemistry and Biophysics, University of California, San Francisco, San Francisco, United States; [7]Institute for Protein Design, University of Washington, Seattle, United States

*For correspondence: dimaio@uw.edu

Present address: [†]Department of Biochemistry and Biophysics, University of California, San Francisco, San Francisco, United States; [‡]Cyrus Biotechnology, Seattle, United States

**Abstract** Cryo-EM has revealed the structures of many challenging yet exciting macromolecular assemblies at near-atomic resolution (3–4.5Å), providing biological phenomena with molecular descriptions. However, at these resolutions, accurately positioning individual atoms remains challenging and error-prone. Manually refining thousands of amino acids – typical in a macromolecular assembly – is tedious and time-consuming. We present an automated method that can improve the atomic details in models that are manually built in near-atomic-resolution cryo-EM maps. Applying the method to three systems recently solved by cryo-EM, we are able to improve model geometry while maintaining the fit-to-density. Backbone placement errors are automatically detected and corrected, and the refinement shows a large radius of convergence. The results demonstrate that the method is amenable to structures with symmetry, of very large size, and containing RNA as well as covalently bound ligands. The method should streamline the cryo-EM structure determination process, providing accurate and unbiased atomic structure interpretation of such maps.

## Introduction

Advances in direct electron detectors, as well as better image analysis algorithms, have led cryo-electron microscopy (cryo-EM) to achieve near-atomic resolution (3–4.5 Å) using single-particle analysis (*Li et al., 2013*; *Bai et al., 2013*; *Scheres, 2012*). Cryo-EM reconstructions at these resolutions, at which individual β-strands are resolvable and bulky sidechains are somewhat visible, make it possible to build an all-atom model directly from such maps (*Kudryashev et al., 2015*; *Wang et al., 2015*). Although sequence can be registered, density maps at this range of resolution do not grant enough information to assign coordinates for each atom in the structure precisely, and thus they cannot always capture molecular interactions for a biochemical process. Furthermore, such model building and refinement is challenging and error prone (*DeLaBarre and Brunger, 2006*; *Brunger et al., 2009*). Nevertheless, determination of detailed atomic interactions from these sparse sources of

data is desirable. However, the inherent ambiguity in the data makes accurately identifying molecular interactions extremely difficult, even for experts.

Model-building into a cryo-EM map at near-atomic resolution generally involves manually building a model into the map with the assistance of a graphical user interface tool (*Emsley et al., 2010*), followed by refinement with software repurposed from X-ray crystallography (*Brown et al., 2015*; *Afonine et al., 2012*). This process requires identification of key amino acid sidechains to register stretches of sequence within the map (possibly aided by information on the topology of a homologous structure), followed by extension of these short fragments of sequence to form one or more fully connected protein chains. At near-atomic resolution, this manual model-building and refinement can be error prone because: (a) the density may not be of sufficient resolution to identify sidechain rotamers uniquely, even for bulky aromatic residues, making it difficult to determine sidechain-sidechain or sidechain-backbone interactions accurately; (b) for regions of non-regular secondary structure (turns or loops) or with poor local resolution, it may be difficult to position backbone atoms accurately; and (c) in these same regions, precise sequence registration may also be error prone. Getting these atomic interactions correct is crucial for understanding detailed atomic mechanisms of protein activities, for designing drugs with a very specific shape complementarity, and for understanding subtle conformational changes of a protein. A structure refinement procedure that can automatically improve the atomic details of a model from such density data is thus very much desired.

In this manuscript, we develop a three-stage approach for automatically refining manually traced cryo-EM models (*Figure 1*). While previously we have developed an iterative local rebuilding tool that is capable of refining homology models into near-atomic-resolution cryo-EM maps (*DiMaio et al., 2015*), several advances were required to extend this tool so that it can successfully refine hand-built models. Our new approach includes a method for automatically detecting and correcting problematic residues in hand-built models without overfitting, a model-selection method for identifying models with good agreement to the density data and with physically realistic geometry, a voxel-size refinement method for correcting errors in calibrating the magnification scaling factor of a microscope, an improved sidechain-optimization method to correct sidechain placement errors in very large systems, and a way to estimate uncertainty in a refined model. These methods, combined, allow the tool to correct backbone errors that significantly deviate from the starting model, but it may still assign a high degree of confidence to these regions in the refined model.

Finally, we apply this approach to three recently solved cryo-EM single particle reconstructions at near-atomic resolution: the TRPV1 channel at 3.4-Å resolution (TRPV1) (*Liao et al., 2013*), the $F_{420}$-reducing [NiFe] hydrogenase (Frh) at 3.4-Å resolution (*Allegretti et al., 2014*), and the large subunit of mitochondrial ribosomes at 3.4-Å resolution (mitoribosome) (*Brown et al., 2014*). We show that in all three cases of diverse and large systems, we are able to refine models automatically to high-quality (as assessed by MolProbity), while maintaining or improving agreement to the density data. Significantly, in the case of TRPV1, we newly identify a biologically relevant atomic interaction – a disulfide bond – that was not built in the originally deposited model but that is supported in the literature. In the case of Frh, we show that our refinement procedure led to a significant improvement of model geometry. Finally, in the case of mitoribosome, we show significant improvement in model geometry: the number of 'Ramachandran favored' residues increases by 5%, and Molprobity score (*Chen et al., 2010*) improvement is observed in all 48 protein chains.

## Results

An overview of our refinement approach is shown schematically in *Figure 1* (and is fully described in Materials and methods). Broadly, the approach proceeds in three stages. In the first stage, we identify problematic residues by assessing local model-strain and local agreement to density data. These problematic regions are rebuilt against a 'training' half-map using fragment-based Monte Carlo sampling with many independent trajectories followed by all-atom refinement. Second, the best subset of these independent trajectories are selected by identifying a subset of stereochemically correct models with best agreement to an independent 'validation' half-map, to prevent overfitting. Finally, models are further optimized in the full-reconstruction with a weight optimally scaled between experimental data and the forcefield using the 'validation' half map. Our approach adopts and improves upon our previous work on refining cryo-EM structures from distant homology structures

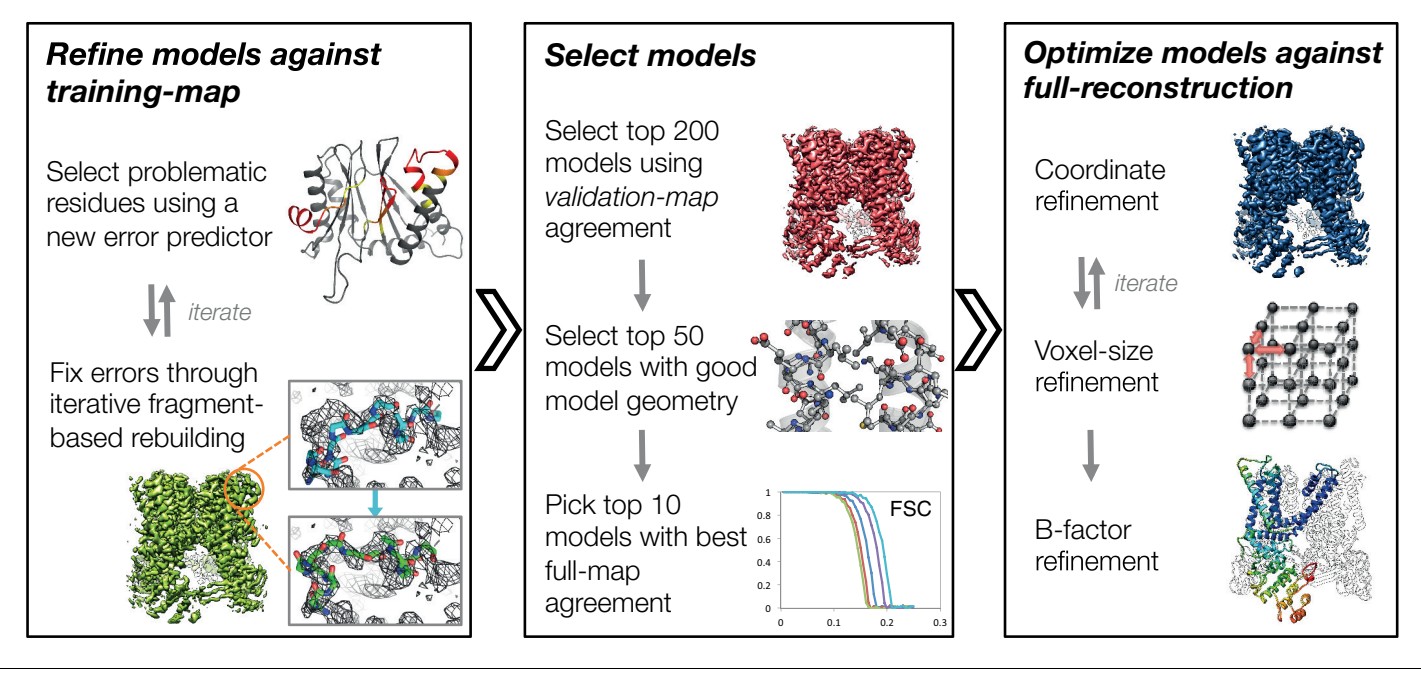

**Figure 1.** An overview of the three stages of automated refinement. (Left) In stage 1, problematic regions are predicted using a newly developed error predictor that looks for local strain in the model and poor local density-fit. These selected regions are subject to iterative fragment-based rebuilding within a Monte Carlo sampling trajectory. Refinement in this stage is restricted to using one-half of the data, referred to as the training map. (Middle) In stage 2, the best models from the ~5000 independent Monte Carlo trajectories are selected. Models are selected based: on agreement to the validation map (independently constructed from the other half of the data), then by model geometry as assessed by MolProbity, and finally, on agreement to the full reconstruction. At this point, the selected models should in general have good fit-to-density and good geometry without overfitting to the data. (Right) In stage 3, using the 10 best models selected, we then optimize against the full reconstruction. Two half maps are used to choose the optimal density weight to refine structures using full-reconstruction. Finally, these top 10 models are optimized (without large-scale backbone rebuilding) into the full-reconstruction, which alternates with voxel-size refinement iteratively. Finally, these models are subject to B-factor refinement.

The following figure supplements are available for figure 1:

**Figure supplement 1.** A close-up view of model strain indicating errors in density-optimized TRPV1 models using the superceded Rosetta approach.

**Figure supplement 2.** Incorporating model strain improves error detection.

**Figure supplement 3.** Density weight optimization against half maps for Mitoribosome.

**Figure supplement 4.** Model geometry is improved with a separate pre-proline potential.

(*DiMaio et al., 2015*), in which a similar fragment-based backbone rebuilding strategy was employed. However, several crucial improvements were necessary in extending our previous work to successfully refine hand-traced models, larger complexes, and a more diverse set of systems.

## Identification of backbone errors using local strain

In our previous work (*DiMaio et al., 2015*), local fit to density is used to identify residues in a distant homology model. Unlike remote homology models, hand-traced models typically fit the data very well, but are incorrect geometrically (in terms of strain). Consequently, a key improvement is to make use of model strain as a criterion in selecting regions to refine. Moreover, when this previous approach was applied to the *de novo* hand-traced models from cryo-EM maps, we observed that in incorrect regions, the models still fit the density well, but did so by introducing strain into the nearby bond angles and torsions. This often occurred near the Cβ atom of aromatic residues, where strain

was introduced to fit the sidechain into the density (*Figure 1—figure supplement 1*). We reasoned that the backbone was incorrect in these strained residues; by correcting the backbone, we would be able to fit a non-strained sidechain into density. Thus, local strain can serve as an indicator to identify regions that could be refined to improve both the fit-to-density and the model geometry. We developed an error predictor by constructing a function (see Materials and methods) that assesses both local model-map agreement and local model-strain. Using a training dataset composed of error-containing models of a cryo-EM map in which the structure has been determined by X-ray crystallography (*Figure 1—figure supplement 2*), we show that the new error predictor offers better discrimination of incorrectly versus correctly placed backbone, with an AUPRC (area under precision-recall curve) of 0.80 versus 0.76 using density alone (*Figure 1—figure supplement 2*). In cases where models are hand-built into density, we expect this strain term to play an even larger role, as fit-to-data is expected to have larger influence on the initially constructed model.

## Better treatment of sidechain density

Recent studies have shown that certain sidechains – particularly those containing negatively charged amino acids (Glu/Asp) – tend to suffer from radiation damage and thus appear weaker in single-particle reconstructions (*Bartesaghi et al., 2014*; *Campbell et al., 2015*). Moreover, the density of certain bulky sidechains, such as those containing Lys and Arg, tends to be less well-defined than their backbone density. This missing density dramatically affects the convergence of conformational sampling during structure refinement, during which sidechains tend to be fit into density corresponding to backbone atoms. To compensate for this, we downweigh the contributions of sidechains which are less resolved in cryo-EM density. Down-weighing factors for each amino acid were determined by comparing the average per-amino-acid real-space B-factor on two cryo-EM reconstructions with known high-resolution crystal structures (20S proteasome [*Li et al., 2013*] and β-galactosidase [*Bartesaghi et al., 2014*]), with the ratio of backbone and sidechain average B-factors being used to derive the scaling factors. Table 3 shows the computed scalefactors used in our refinement method.

## Local sidechain refinement for large complexes

When our previous all-atom refinement approach was applied to very large complexes (of 800+ residues), we observed many instances in which sidechains were not properly optimized to density (*Figure 6—figure supplement 1*). It was hypothesized that this was due to the convergence of sidechain optimization, as the number of possible sidechain states expands exponentially with the number of residues present in a protein. Here, we opted to treat this global optimization problem as a series of smaller local optimization problems, repeatedly optimizing overlapping regions of ~20–100 residues until all residues in a protein are visited at least once. This approach resolved this sidechain fitting issue, as shown in *Figure 6—figure supplement 1* (right panel).

## Voxel-size refinement

The voxel size of a cryo-EM reconstruction is determined by the physical pixel size on the detector scaled by a magnification factor. However, the magnification factor may be determined with some inaccuracy, leading to errors in deciding the voxel size of the resulting single-particle reconstruction. It has been shown that voxel size may be off by as much as several percent when using EM maps as molecular replacement targets (*Jackson et al., 2015*). Here, we develop a voxel-size refinement strategy that scales the voxel size of the map to maximize the model-map real-space correlation coefficient. During refinement, we alternate structure refinement and map voxel size refinement with several cycles iteratively until the voxel size converges (*Figure 1*). The approach is fully described in the Materials and methods section.

Moreover, we investigate the robustness of our voxel-size refinement method in the presence of model errors, and demonstrate that our iterative approach captures the general agreement between forcefield and voxel size. We initially made use of an arbitrary target structure (PDB id: 4AKE). We calculated density to 3Å resolution on a 1Å grid and ran several MD trajectories in Rosetta, followed by all-atom minimization, yielding 50 models that are 2.9–3.1Å RMSd from the native structure. We initially refined voxel size against each of these models, yielding voxel sizes from 0.95 to 1.02 Å (stdev = 0.011). Following our iterative procedure, the deviation was much smaller, ranging from 0.99 to 1.02 (stdev = 0.004). *Figure 2A* illustrates the distribution of voxel sizes derived from the

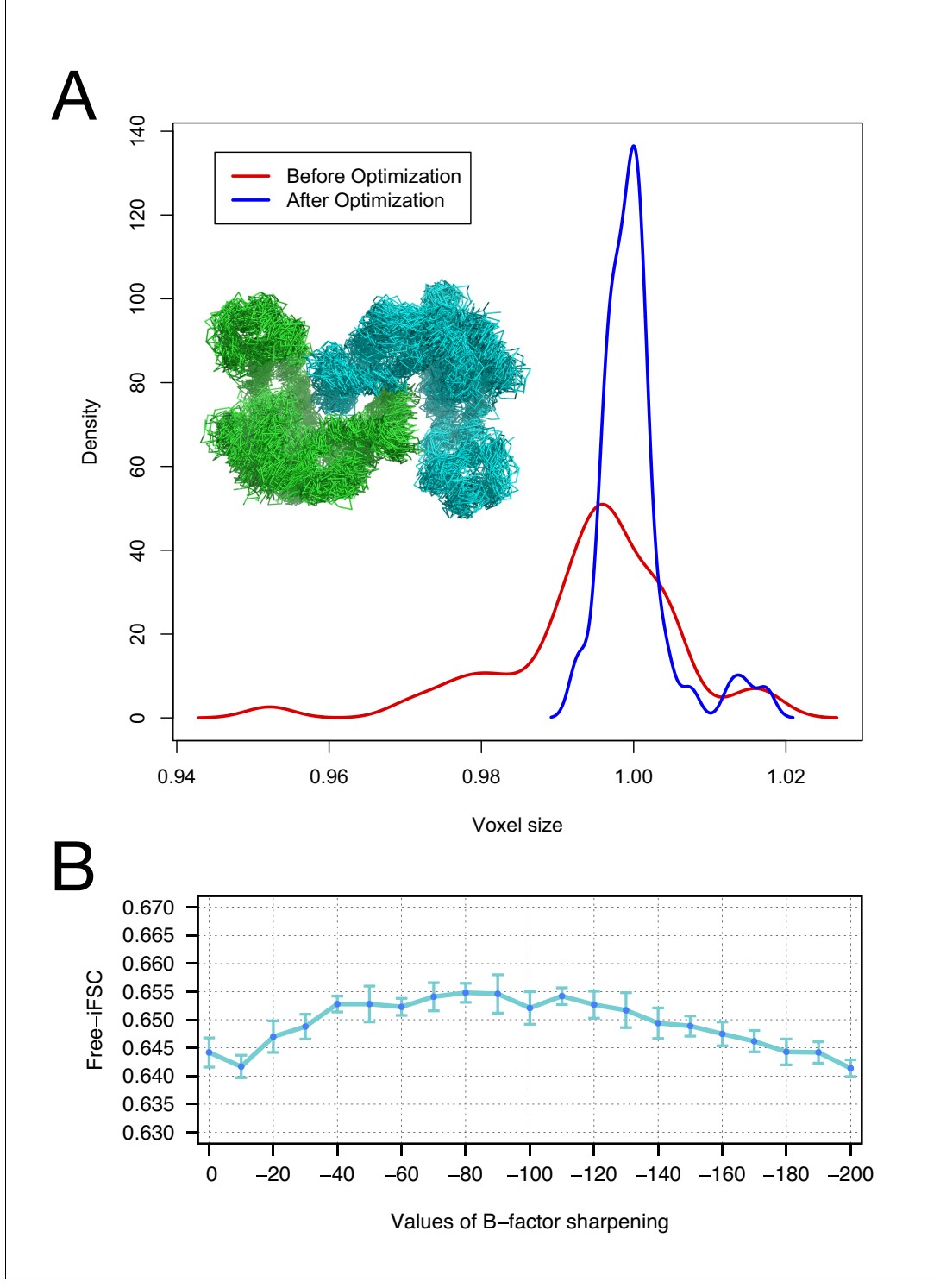

**Figure 2.** The accuracy of voxel size refinement and the effect of B-factor sharpening in Rosetta refinement. (**A**) Voxel-size refinement on perturbed models. Perturbed structures were generated by running short MD trajectories in Rosetta, followed by all-atom minimization. Voxel size is refined against the perturbed models, yielding the density distribution in red. Following cycles of iterated voxel refinement and all-atom refinement, the voxel size shows significantly better convergence (blue line). (**B**) Rosetta structure refinement with a range values of B-factor sharpening. Integrated Fourier Shell Correlation eavluated using the validation map (free-iFSC) is plotted here as a function of B-factor sharpening of the training map. The results indicate that our refinement method is not particularly sensitive to the extent of B-factor sharpening, behaving similarly over a range of sharpening values

*Figure 2 continued on next page*

*Figure 2 continued*
between −40 and −130. The error bars show standard deviation of the free-iFSC among the top10 ensemble models (see Materials and methods for the ensemble selection method).

models before refinement (red curve) and after refinement (blue curve), showing a sharp peak at the true voxel size (1.0Å) after refinement.

## The effect of B-factor sharpening in Rosetta structure refinement

To investigate the extent of B-factor sharpening that would influence the accuracy of Rosetta structure refinement, we benchmarked structure refinement using various B-factors from 0 to −200 (*Figure 2B*). Using the 20S proteasome 3.3Å resolution map, we refined a model starting from a template (PDB id: 3H4P, 52% sequence identity) in the training map, and reported the integrated Fourier Shell Correlation evaluated by the validation map (free-iFSC). The method does not show a particularly strong dependency on B-factor sharpening values; with B-factors ranging from –40 to –130, the refinement all performed equally well, as assessed by free-iFSC.

## The role of independent reconstruction in Rosetta structure refinement

In our previous approaches, we used independent reconstructions ('validation' half-map) both for model selection (*DiMaio et al., 2015*) and to determine the balance between model geometry and fit-to-data during refinement (*DiMaio et al., 2013*). In this manuscript, we use independent reconstructions in the same manner during the first two stages of refinement (*Figure 1*). At the very last stage, however, we perform several steps in the context of the full reconstruction because of the additional sidechain details that may be present only in the full reconstruction. As shown in *Figure 1*, for the best 10 sampled models selected from stage 2, we perform a final all-atom and atomic B-factor refinement against the complete reconstruction. Similar to the approach adapted by the REFMAC group (*Brown et al., 2015*; *Fernández et al., 2014*), we use two independent halves of the data (the training and validation half-maps) to optimize the weight used with full-reconstruction data (see Materials and methods and (*Figure 1—figure supplement 3*), and that weight is used both in refinement against the full reconstruction and in voxel-size refinement. Following refinement against the full reconstruction, model geometry is verified (using MolProbity [*Chen et al., 2010*]) to ensure that it is not worsening during refinement against the full reconstruction. This confers additional sensitivity during model selection.

## Evaluation of refined models with Molprobity and EMRinger

Models are evaluated for geometric quality using Molprobity (*Chen et al., 2010*), which compares local geometric features describing an all-atom model to those from high-resolution crystal structures. In addition to using MolProbity to assess model quality, we further validate the Rosetta-refined models with EMRinger (*Barad et al., 2015*), using this tool as an independent source to validate both model geometry and density-fit at sidechain level. EMRinger samples density around Cγ atoms as they are rotated about the $\chi_1$ dihedral angle, and identifies the angle which presents peak density for the Cγ. On the basis of existing statistical and chemical information, this position should generally fall into the rotamer distribution of $\chi_1$, with angles of 60, 180, and 300 degrees. The distribution of measured peak angles at various signal-to-noise cutoffs is integrated into the EMRinger score, which reports on backbone model-to-map agreement using side chain geometry.

## Application to TRPV1

We first applied our new refinement approach to the recently determined 3.4-Å cryo-EM reconstruction of the TRPV1 channel in the apo form (*Liao et al., 2013*). Half-maps were reconstructed by subdividing particles into two sets randomly, with one used for initial model rebuilding and refinement, and the other used for validation. The deposited model (PDB id: 3J5P) was used as input to the protocol described previously. All refinement was carried out using the native C4 symmetry. Because of the highly heterogeneous nature of the system, fragment-based rebuilding (stage 1 and 2) was focused on the trans-membrane domain initially, and then the full three-stage refinement was

performed on the full structure (see Materials and methods for details). All input files are included as *Supplementary file 1*.

The results of refinement are indicated in *Figures 3* and *4* and in *Table 1*. The refined model improves model quality and maintains good model-data agreement when compared to the deposited model: the MolProbity score improves from 3.81 to 1.45, the fit-to-data (integrated Fourier shell correlation from 10 to 3.4Å) drops slightly from 0.612 to 0.607, but the EMRinger score improves from 0.65 to 2.34, indicating that the better fitting shown in the deposited model might result from overfitting. *Figure 3A–B* compares the refined model and the deposited model, illustrating the model violations reported by MolProbity. *Figure 4A* illustrates the convergence of our refined ensemble, showing the 10 selected refined structures, the top model colored by per residue structural variation, and the refined B-factors. Both structural variation and B-factors provide unique insights on assessing the local confidence of the refined models, in which structural variance shows the allowed local conformations that satisfy the density data, while B-factors assess the local resolution of the density data at different regions of a model.

Closer inspection of the low-energy refined models identified that serveral made a disulfide linkage (C386–C390) that was not built in the deposited model and was supported by the deposited density data (*Figure 4B*). Further refinement (with the addition of the Ankyrin repeat domain) was made using the lowest energy disulfide-containing model as a starting point. This disulfide has previously been identified and characterized in the literature as playing an important role in the TRPV1 channel's response to oxidative stress (*Chuang and Lin, 2009*); this, combined with our model's ability to better explain a tube of density that is unaccounted for in the deposited model, let us speculate that this disulfide bond is present in the cryo-EM reconstruction. The discovery of this linkage also illustrates the magnitude of conformational change that may be captured by our protocol; our Monte Carlo backbone sampling strategy allows refinement to overcome energy barriers that other methods using density minimization alone cannot overcome. Despite the magnitude of these changes, the conformational ensemble is well converged in this region (*Figure 4B*, right panel), providing further confidence in our refined model.

## Refinement of highly liganded complexes: application to the $F_{420}$-reducing [NiFe] hydrogenase complex

As our next test of the approach, we wanted to illustrate structure refinement of a macromolecular assembly containing proteins with large numbers of ligands, some of which are covalently bound, all in a system with high-order point symmetry. For this, we chose the 3.4-Å reconstruction of the $F_{420}$-reducing [NiFe] hydrogenase complex, in which the asymmetric unit contains three protein chains which feature together with a [NiFe] cluster, two metal ions, four [4Fe4S] clusters that are covalently bound to cysteine sidechains, and a flavin adenine dinucleotide (FAD) (*Figure 5A*). The complex is a dodecamer with tetrahedral symmetry, with 12 copies of a 902-residue molecule of three protein chains. We used the -*auto_setup_metals* option of Rosetta to maintain covalent linkages between protein and ligand during refinement (full input files are included as *Supplementary file 2*). The results of refinement are indicated in *Figure 5* and *Table 1*; the MolProbity score improves from 3.98 to 1.59, and the EMRinger score improves from 1.06 to 2.17, but the iFSC drops from 0.743 to 0.708. We reason that the decrease of fit-to-density at high-resolution (10–3.4Å) shells may be a result of overfitting the model to the density map; the deposited model was forced to fit the density by deviating the likely geometry observed in high-resolution crystal structures. This overfitting hypothesis is well supported by a high number of bad clashes and 39% rotamer outliers found in the deposited model (*Table 1* and *Figure 5C*).

## Refinement of large complexes: application to the mitochondrial ribosome large subunit

Finally, we wanted to test the ability of our refinement to scale to large asymmetric macromolecular assemblies, more typical of cryo-EM single particle reconstruction. To do so, we considered refining models against the previously published 3.4-Å cryo-EM reconstruction of the large subunit of the human mitochondrial ribosome (*Brown et al., 2014*). The deposited model had been previously refined with REFMAC (*Brown et al., 2015*), and consists of 48 chains of proteins with 7469 amino acids assigned, and two chains of RNA with 1529 nucleic acid bases.

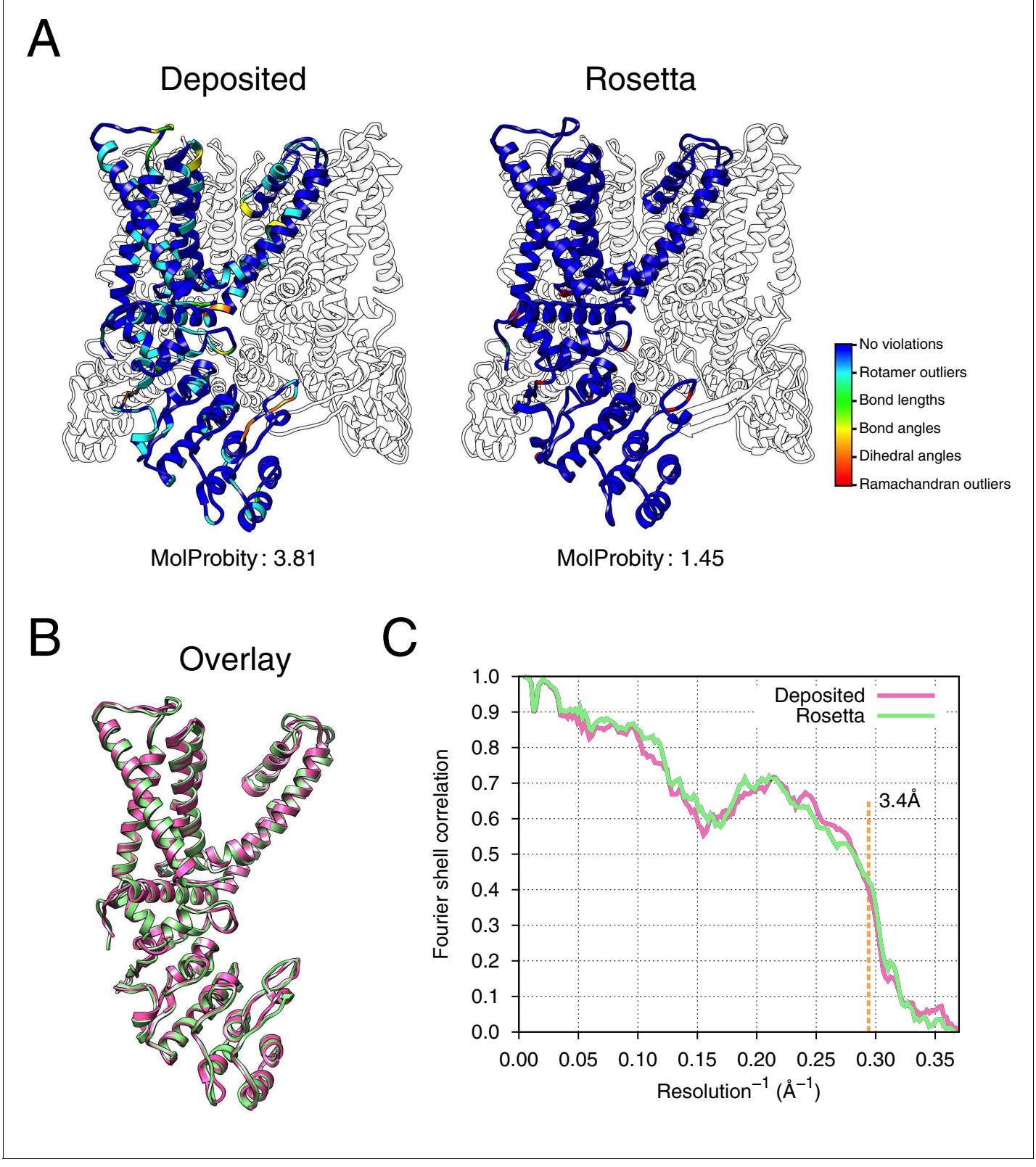

**Figure 3.** Refinement of the apo TRPV1 channel (EMD-5778) shows improved model quality. (**A**) Comparison of the deposited and Rosetta-refined models, as assessed by MolProbity. Residues reported as violations are colored using the key shown on the far right. Blue open arrows indicate that the hydrogen-bond geometry of a β-hairpin was automatically detected and improved in the Rosetta refined model. (**B**) An overlay of the asymmetric unit of the deposited (pink) and the Rosetta-refined (green) model indicates the magnitude of conformational changes that are explored by our

*Figure 3 continued on next page*

*Figure 3 continued*

refinement approach. (**C**) The agreement of models to map assessed by Fourier space correlation (y-axis) at each resolution shell (x-axis), where the reported resolution (3.4Å) is depicted in a dashed orange line.

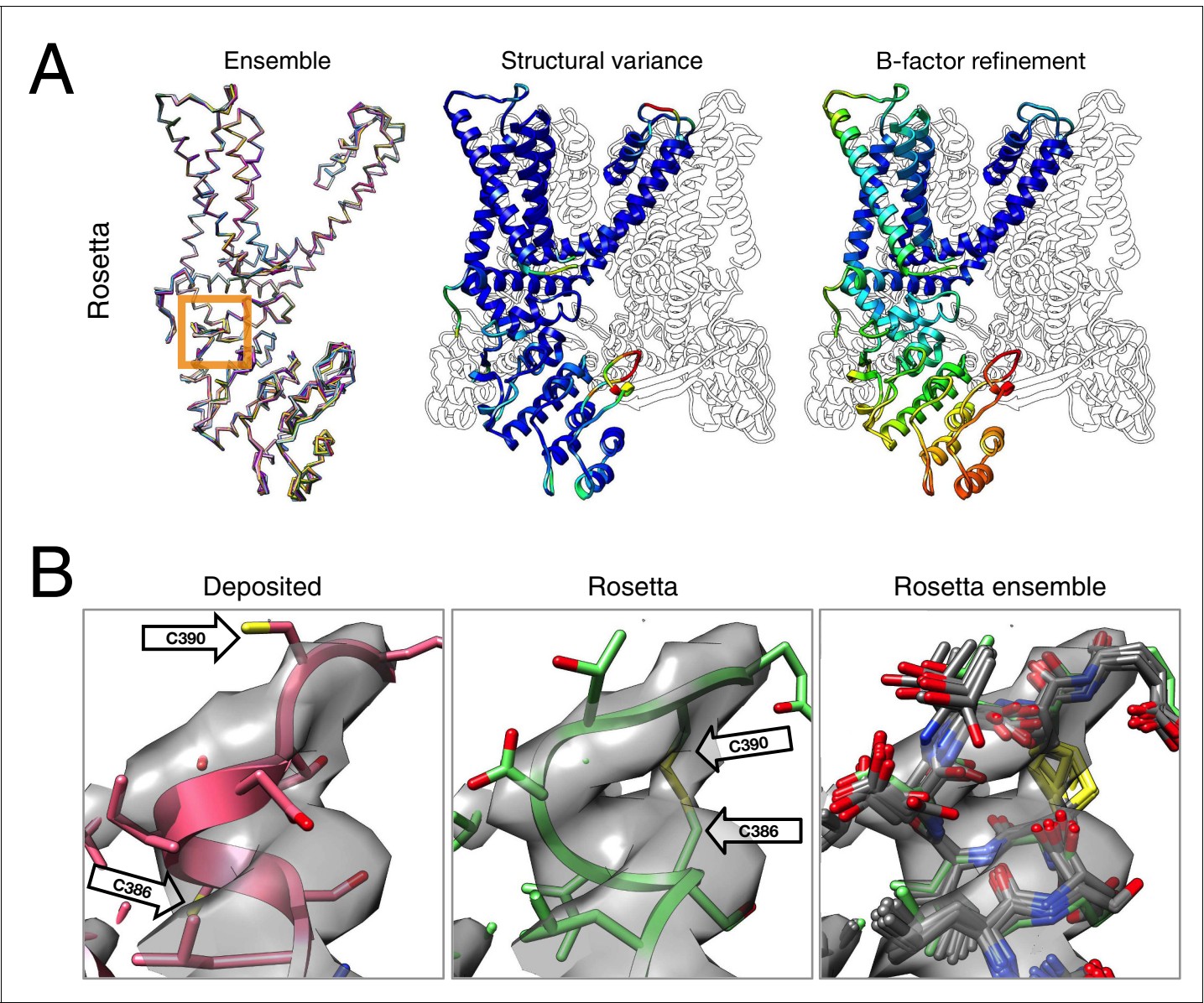

**Figure 4.** Refinement of the TRPV1 channel identifies a previously unmodeled disulfide bond. (**A**) An overview of the entire structure, estimating local model uncertainty in two ways: local structural diversity and refined B-factors. Local structure diversity is indicated by showing (left) an overlay of the top 10 Rosetta models, (middle) the top model colored by per residue deviation, and (right) the refined per-atom B-factors. Using the model selection method illustrated in the middle panel of *Figure 1*, the Cα RMSDs among the selected ensemble range from 0.44 to 0.63 Å. The orange square shows the location of a newly identified disulfide bond (C386–C390) revealed by our refinement protocol. (**B**) A zoomed-in view of the disulfide linkage (C386–C390) identified by the automated method. Note that the sidechain coordinates of C390 were unassigned in the deposited model; for presentation, the sidechain atoms of C390 were optimally added by Rosetta on the basis of the deposited backbone torsion angles of C390.

**Table 1.** Structure refinement of macromolecular assemblies from cryo-EM maps using Rosetta.

| | EMD ID | PDB ID | Reported resolution [Å] | Symmetry | Number of amino acids* | MolProbity[†] | | | | EMRinger score[†] | iFSC[‡] |
|---|---|---|---|---|---|---|---|---|---|---|---|
| | | | | | | Score | Clash score | Rotamer outliers [%] | Ramachandran favored [%] | | |
| TRPV1 | 5778 | 3j5p | 3.4 | C4 | 489 (1956) | 3.81 / 1.45 | 86.35 / 1.96 | 28.78 / 0.00 | 95.65 / 91.93 | 0.65 / 2.34 | 0.612 / 0.607 |
| Frh | 2513 | 4ci0 | 3.4 | T | 893 (10,716)[§] | 3.98 / 1.59 | 120.42 / 3.22 | 39.11 / 0.27 | 96.51 / 92.18 | 1.06 / 2.17 | 0.743 / 0.708 |
| Mitoribosome | 2762 | 3j7y | 3.4 | N/A | 7469[¶] | 2.71 / 1.50 | 8.38 / 3.51 | 8.49 / 0.08 | 89.86 / 94.86 | 2.09 / 2.40 | 0.692 / 0.676 |

*Number of protein residues in the asymmetric unit and (the total residues) modeled.

[†]Scores from deposited (left) versus (/) Rosetta refined (right) model.

[‡]Integrated Fourier Shell Correlation (iFSC) from 10–3.4Å resolution shells.

[§]In addition to protein residues, nine residues of ligand per asymmetric unit–including a [NiFe] cluster, two metal ions (Fe and Zn), and four [4Fe4S] clusters, and an FAD–were included in the refinement.

[¶]In addition to protein residues, 1529 base pairs of RNA molecule were included in the refinement.

In order to make conformational sampling tractable, we used a strategy slightly modified from that shown in *Figure 1* (full input files are included as *Supplementary file 3*). The first two stages of the protocol (fragment-based backbone rebuilding and model selection) were carried out on each protein chain individually, whereas the third stage was carried out on the fully assembled complex. Model selection was carried out on each individual chain; each selected model was refined in the context of the complete assembly. Nucleic acids were not subject to geometry refinement but were included as rigid bodies throughout the whole-assembly refinement to accurately recapitulate protein–RNA interactions.

The results of the refinement are indicated in *Figure 6* and in *Table 1*. Several large-scale conformational changes again appear in converged models; these models show better geometry, improved fit-to-density and fewer unexplained regions of density. The backbone geometry improvements in particular are noticeable in proteins with β-sheet-containing domains. Refinement procedures such as *phenix.real_space_refine* [*Afonine et al., 2012*] and REFMAC [*Brown et al., 2015*] require manual input of secondary structure restraints that are determined either from an initial model or from homologous protein structure to maintain backbone geometry during refinement. In our approach, the Rosetta forcefield is able to optimize hydrogen bond geometry in secondary structures without requiring prior knowledge of secondary structures. This is particularly powerful in refining *de novo* structures whose secondary structure is ambiguous because of poor local resolution. *Figure 6C* illustrates an example (chain k) of this from the case of mitoribosome, where a β-sheet not present in the original model is identified, the backbone geometry is improved, and the model fits the density much better than the deposited model (*Figure 6B* left panel, red arrow); the refinement also shows a large radius of convergence.

The refined ribosome model has 1.50 MolProbity score, 0.676 iFSC, and 2.40 EMRinger score. The largest improvements tend to occur in regions of low local resolution (~5Å assessed by ResMap [*Kucukelbir et al., 2014*] from the original paper) on the periphery of the complex. Looking at the results on individual chains, the MolProbity score improves on all 48 protein chains, in part because of the much-improved backbone geometry assessed by the *Ramachandran favored* term in MolProbity (*Figure 6A*, right panel). Our Monte Carlo backbone sampling can correct these incorrect backbone placements, which often require significant compensating conformational changes. EMRinger score is also consistently improved (*Figure 6—figure supplement 2*), particularly in regions where the deposited model scores poorly.

## Comparison to *phenix.real_space refinement*

Finally, using the same set of target proteins, we compare the Rosetta refinement results with those of another state-of-the-art real-space refinement method from the *phenix* package (*phenix.real_space_refine*) (*Afonine et al., 2012*). In order to prevent refinement from fitting to noise,

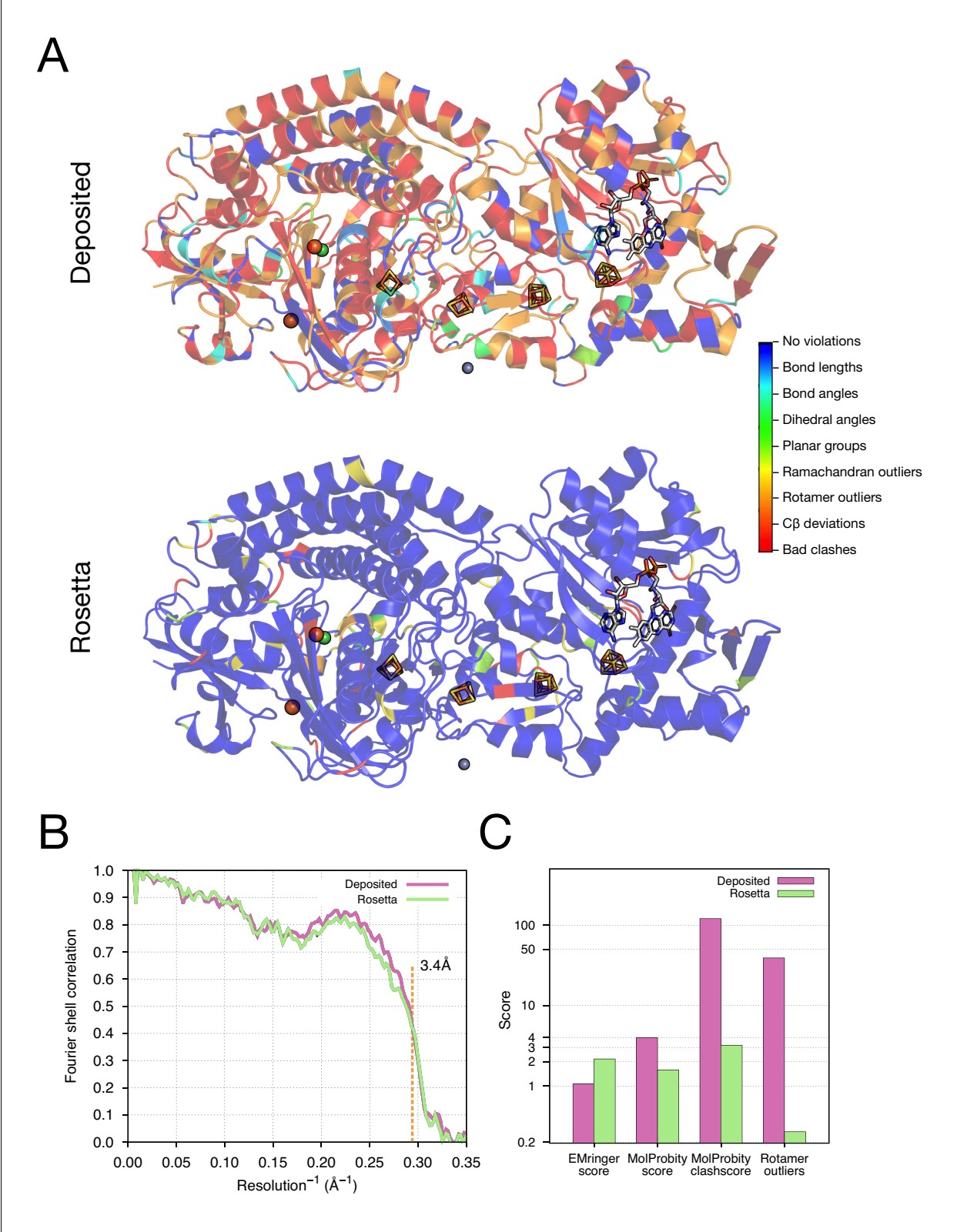

**Figure 5.** Refinement of the $F_{420}$-reducing [NiFe] hydrogenase (EMD-2513) improves the model geometry. (**A**) An illustration comparing the model geometry of the deposited (upper panel) and Rosetta-refined (lower panel) models. Three chains (A/B/C) of the asymmetric unit of the complex are shown as cartoon with geometry violations reported by MolProbity colored according to the key shown on the far right. Four iron–sulfur clusters [4Fe4S] and a FAD are shown in a stick representation. Metal ions are depicted as spheres, with Zn grey, Fe orange, and Ni green. (**B**) Model–map agreement –
*Figure 5 continued on next page*

*Figure 5 continued*

as assessed by Fourier shell correlation (y-axis) as a function of resolution (x-axis) – quantifies this improvement following voxel-size refinement. (C) Model quality as assessed by EMRinger and MolProbity. The x-axis shows methods used to evaluate the models, while the y-axis shows the scores under each criterion.

The following figure supplement is available for figure 5:

**Figure supplement 1.** The symmetry operators denoted in the deposited PDB (PDB 4ci0) produce a complex that could not fit into the deposited density map properly.

starting from the deposited models, we carried out *phenix* real-space refinement in the training maps, which contained only the half of the data used by Rosetta in the first step of the refinement procedure (*Figure 1*). We then used the validation map to evaluate and compare the *phenix* refinement results to the Rosetta-refined models before full-map refinement. In the case of TRPV1, *phenix* used 0.24 CPU hours, generating a single model. For Rosetta, the 1000 independent trajectories take about 5 hr each, or 5000 CPU-hours in total. As shown in *Table 2*, with much shorter run time (at most 1 hr) *phenix* can yield models with geometry almost as good as Rosetta, albeit with slightly worse density fit as evaluated by both real-space correlation coefficient and iFSC. However, without the Monte Carlo backbone conformational sampling used by Rosetta, models generated from *phenix* tend to minimally perturb the structure, and cannot provide the large backbone corrections shown by our new approach.

## Discussion

In this report, we develop a method for improving the atomic details of manually traced models from 3–4.5Å resolution cryo-EM density maps. We show the applicability of the approach by applying it to three systems: a membrane protein, a highly symmetric system with a large number of ligands, and an asymmetric macromolecular assembly containing large numbers of protein chains and RNAs. In all cases, we show that we are able to improve model geometry significantly while maintaining good agreement to the density data. We show that model convergence can be used to suggest local model uncertainty in addition to B-factors. Finally, we also show that our models also recover structure features that are supported in the literature, or that are in much better local agreement with the density data.

Unlike other approaches (*Brown et al., 2015*; *Headd et al., 2012*), our approach can automatically perform large-scale backbone reorganization, correcting backbone placement errors common in these 3–4.5Å resolution datasets. Two features of our refinement approach regarding the use of prior information are crucial to the success of this large-scale refinement. First, the use a physically realistic forcefield throughout refinement handles the under-constrained nature of refinement at these resolutions, using chemical 'domain knowledge' learned from high-resolution crystal structures to implicitly fill in the missing information in the data. Second, our fragment-based rebuilding , which explicitly samples the most likely backbone conformations given a short stretch of sequence, also uses prior information gathered from high-resolution protein structures, further restricting conformation space and filling in additional information that is not present in the data.

In all cases, we found that the high-resolution density-fit (evaluated by FSC integrated from 10–3.4Å resolution shells) drops slightly after Rosetta structure refinement. We reason that this is probably because the deposited models were overfit to the density maps, so that the models (especially the sidechains) were forced to fit into the density by violating the likely geometry observed from known high-resolution crystal structures. The observation of the slightly decreased of fit-to-density but significant improvement of model geometry bolsters the importance of using prior information (such as sidechain rotatmers), as well as of having a refinement scheme to monitor model overfitting.

There is an open question as to how structure refinement can be further improved, particularly as refinement extends to even lower resolutions (worse than 5Å). Enhancing the predicting power of the Rosetta modeling methods is key to pushing the resolution limit of the current refinement method further. This can be achieved by improving: (1) the energy function (forcefield) used in

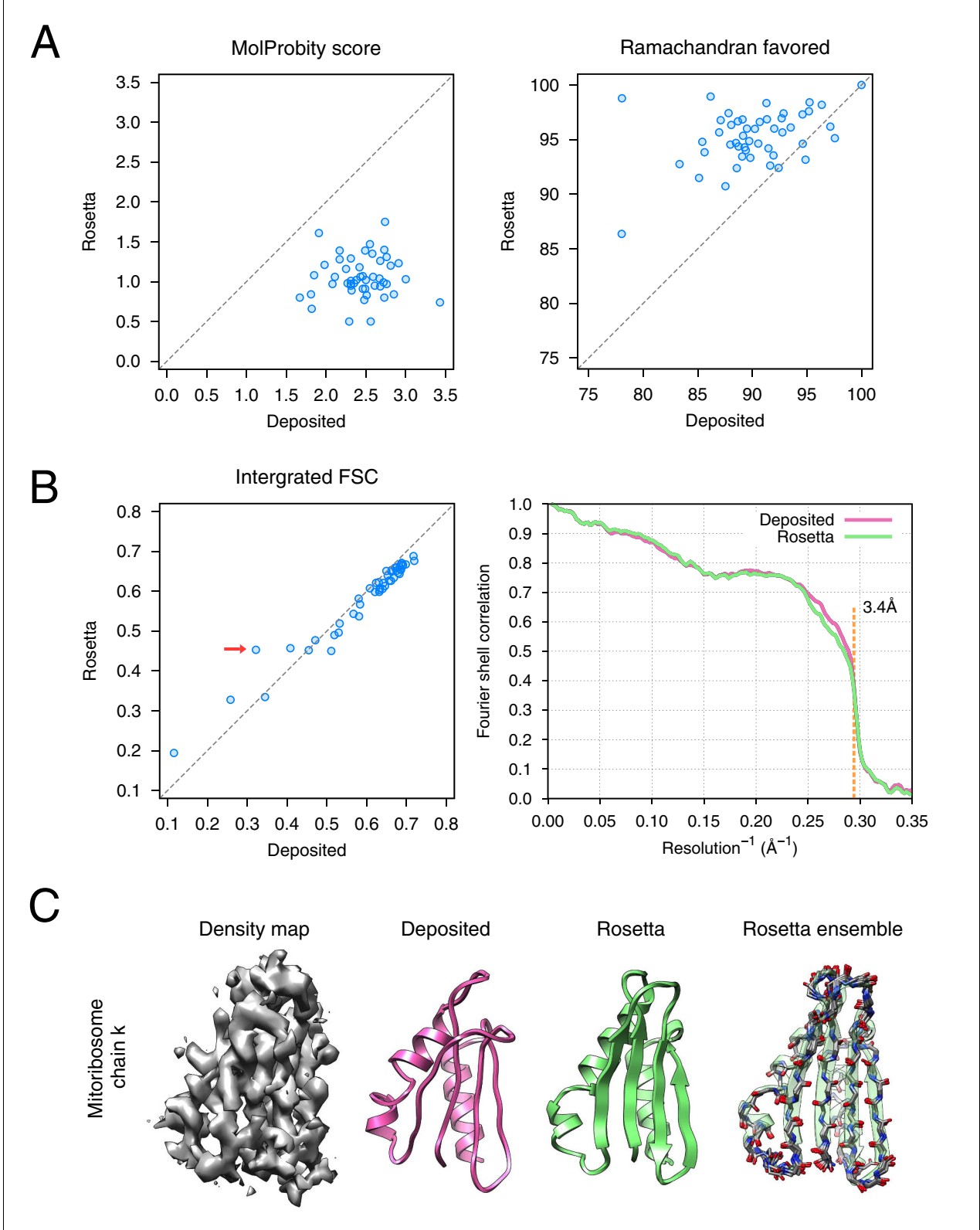

**Figure 6.** Refinement of the large subunit of the human mitochondrial ribosome (EMD-2762) shows improvements to all subunits. (A) Scatterplots of model quality for each of the 48 protein chains compare the deposited (x-axis) and Rosetta (y-axis) models using MolProbity. On the left, the MolProbity scores of all 48 protein chains are compared, where a lower values indicates a better model geometry. On the right, the percentage of 'Ramachandran favored' residues on each chain are compared, with higher values preferable. (B) An evaluation of the fit-to-density of each protein

*Figure 6 continued on next page*

Wang *et al.* eLife 2016;5:e17219. DOI: 10.7554/eLife.17219

*Figure 6 continued*

chain. On the left, we compare the Fourier shell correlation (FSC) of each chain before and after refinement; we integrate the FSC from 10Å to 3.4Å. Higher values indicate better agreement with the data. The largest improvement, chain k, is indicated by the red arrow. On the right, we show the full FSC curve, with the deposited model shown in pink, and the Rosetta refined model shown in green; the reported map resolution (3.4Å) is indicated in the dashed orange line. (C) A zoomed-in view indicating a much improved backbone geometry and the large radius of convergence of the refinement of chain k. The left panel shows that the density for chain k is in the region of relatively low local resolution.

The following source data and figure supplements are available for figure 6:

**Source data 1.** Mitoribosome per-chain refinement results.
**Figure supplement 1.** Local relax shows better placement of sidechains for large systems.
**Figure supplement 2.** EMRinger analysis on refinement of the large subunit of the human mitochondrial ribosome.

structure refinement, and (2) conformational sampling methodology, particularly for systems for which secondary structure prediction is poor. Further improvements in the role of B-factor sharpening and its effect on refinement, as well as better predictors of local model error are necessary. Finally, structure refinement in maps that have highly heterogeneous local resolution remains challenging because a single set of refinement parameters cannot readily be applied at all regions. Methodological improvements that allow the adjustment of parameters based on local map quality will be essential to refine structures accurately from such maps. In our effort to enable automated structure refinement for large macromolecular assemblies, we hope that this method will be a valuable tool for determining atomically accurate structures from near-atomic-resolution cryo-EM data.

# Materials and methods

## Preparing maps for refinement

Split maps were provided by the original authors. One map was randomly chosen for refinement, and the other was used for validation. In all cases, a B-factor of −100 was applied to the map used for refinement using the '*image_handler*' tool in *RELION* (*Scheres, 2012*). The maps were subsequently filtered to the user-refined resolution. In the case of the mitochondrial ribosome, segmented maps were prepared using a custom Rosetta application and the deposited structure to guide segmentation:

**Table 2.** Comparison of structure refinement results between Rosetta and phenix.real_space_refine*.

| | RSCC*,†,‡ validation map | iFSC*,†,§ validation map | EMRinger Score*,† validation map | MolProbity† | | | | Number of residues with better RSCC†,¶ |
| | | | | Score | Clash score | Rotamer outliers [%] | Ramachandran favored [%] | |
|---|---|---|---|---|---|---|---|---|
| TRPV1 | 0.785 / 0.790 | 0.546 / 0.566 | 1.84 / 1.90 | 1.59 / 1.48 | 4.30 / 2.14 | 0.00 / 0.00 | 94.41 / 91.72 | 86 / 250 |
| Frh | 0.835 / 0.835 | 0.504 / 0.517 | 1.36 / 1.27 | 1.68 / 1.62 | 7.99 / 3.66 | 0.68 / 0.13 | 96.31 / 92.67 | 677 / 1328 |
| Mitoribosome | 0.832 / 0.832 | 0.476 / 0.478 | 2.05 / 1.98 | 1.88 / 1.62 | 6.17 / 4.08 | 0.38 / 0.00 | 90.19 / 93.49 | 415 / 564 |

*To avoid over-fitting, refinement using both methods was carried out using the half-map approach, in which the models were subject to refinement using the training maps. The results showing here were evaluated using the validation-maps. The input model information is the same as reported at **Table 1**.

†Numbers (scores) from *phenix.real_space_refine* (left) versus (/) Rosetta refined (right) model.

‡Real-space correlation coefficients were evaluated using UCSF Chimera.

§Integrated Fourier shell correlation (iFSC) from 10–3.4Å resolution shells.

¶We calculate per-residue real-space correlation coefficient and report the number of residues which show the value of ΔRSCC greater than 0.05.

*density_tools.default.linuxgccrelease -s 3j7y0.pdb -mapfile EMD-2762.mrc -mask_radius 2 - maskonly*

Some steps of the protocol also made use of the full reconstruction. As with the training map, these were sharpened using a B-factor of –90 with a low-pass filter to 3Å.

## Preparing structures for refinement

In the case of TRPV1, residues 111–202 in the Ankryin repeat domain from the deposited model did not have visible density, and so were deleted prior to refinement. Furthermore, automatic refinement was applied in two stages because of the high heterogeneity between the trans-membrane domain and the Ankyrin repeat domain. The trans-membrane domain (residue 234–586) was first refined in the masked density using the deposited model. In the case of the mitoribosome, residues from chain t and chain f, in which atoms are assigned to residues 'UNK', were removed from all the refinement processes, as well as from data analyses or results comparisons. In the case of Frh, we found that the deposited symmetry operators (the 'BIOMT' lines) are not able to generate a symmetric model that can properly fit into the density map (*Figure 5—figure supplement 1*). To assemble the symmetric complex, we manually docked each subunit of the symmetric complex into the deposited density map (EMD-2513) using Chimera, and used this model as the 'deposited model' used in the paper. The refinement of ligands received special treatment: refinement started using protein only, with constraints maintaining ligand site geometry. Later, ligands were added back on to the protein, which was rerefined.

## Algorithm for model rebuilding

Model rebuilding generally follows the procedure from our previous work (*DiMaio et al., 2015*), with a few key changes highlighted below. Rebuilding starts from the deposited structure, which is first conservatively refined using one macrocycle of the Rosetta *relax* protocol to trigger local strain on sidechains, which iterates four cycles Monte Carlo rotamer optimization with all-atom minimization, ramping the weight on van der Waals repulsion in each cycle. Minimization is carried out in Cartesian space, with a term enforcing ideal bond angles, bond lengths, and planarity (*Conway et al., 2014*).

Following Cartesian minimization, the worst residues are selected using the following equation to evaluate the quality of the model at residue *i*:

$$Z_{error}^{(i)} = w_{dens} \cdot Z_{dens}^{(i)} + w_{lcldens} \cdot Z_{lcldens}^{(i)} + w_{bonded} \cdot Z_{bonded}^{(i)} + w_{rama} \cdot Z_{rama}^{(i)}$$

Four different terms appear in this equation, two of which assess a model's agreement to data, and two of which assess a model's local strain. The first two, $Z_{dens}^{(i)}$ and $Z_{lcldens}^{(i)}$, assess the model–map agreement of the backbone and sidechain atoms of each residue, computing the real-space correlation coefficient in a region around a residue, and converting that to a *Z-score* compared to the entire model. For the former term, an absolute correlation coefficient is computed; for the latter term, the correlation is normalized with respect to residues nearby (those within 10 Å of residue *i*). The latter term is specifically added to deal with maps that have significant diversity in local resolution.

The second two terms, $Z_{bonded}^{(i)}$ and $Z_{rama}^{(i)}$, assess a model's strain following model refinement. The motivation for these terms is that in cases where the model was built incorrectly into density, it will be energetically unfavorable. Following an initial refinement, these incorrect portions will either be moved away from the data, or will introduce model strain to maintain the favorable agreement with the data, depending upon the balance of forces between the two. These terms compare the per-residue bond geometry term, and the per-residue Ramachandran energy, respectively, to that over the entire structure, and compute a *Z-score* for each residue.

For each of the four terms, a *Z-score* is computed and is summed together, with a particular weight for each term. The weights were tuned using a 3.3-Å cryo-EM map dataset with known high-resolution structure (the 20S proteasome [*Li et al., 2013*]), where a set of ~500 error-containing models was used as the training data. The results of this tuning process are shown in *Figure 1—figure supplement 2*. The final weights selected were $w_{dens}$=0.45, $w_{lcldens}$=0.05, $w_{bonded}$=0.15, $w_{rama}$=0.35.

After computing this weighted *Z-score* for each residue, all residues with a score below some target value (see the next section on iteration for specific values) are selected for local rebuilding. Local rebuilding uses the iterative fragment-based approach previously published (*DiMaio et al., 2015*). In our new approach, a residue is randomly chosen from the pool tagged for rebuilding from the previous step. Given the local sequence around this selected residue, a set of 25 protein backbone conformations from high-resolution structures with similar local sequence and predicted secondary structure is sampled. Each sampled backbone is refined – as an isolated fragment – into density using the following three-step procedure: (a) only the backbone is minimized in torsion space using a simplified energy function, (b) sidechain rotamers are optimized into density, and (c) both backbone and sidechain are minimized in torsion space using a simplified energy function. Constraints on the ends of each fragment ensure the a local region is reasonable in the context of the entire backbone. Of the 25 sampled fragments, the best is selected by fit-to-density. Finally, the replaced fragment is minimized in the context of the complete structure. This process is run as a Monte Carlo trajectory.

## Iterative rebuilding and all-atom refinement

Model rebuilding and all-atom refinement are run iteratively, as shown in *Figure 1*. Four separate 200-step Monte Carlo trajectories are run with increasing coverage of predicting errors but sacrificing the accuracy of the predictions. This is done with the *Z-score* cutoff increased in each step, following the schedule shown in *Figure 1—figure supplement 2*: first residues with $Z<-0.5$ are selected for fragment-based rebuilding, followed by $Z<-0.3$, $Z<-0.1$, and finally $Z<0$. Between each cycle, a single iteration of *Relax* is run, in the same manner as in the pre-refinement step. At the start of each stage, the $Z_{error}^{(i)}$ of a model is re-evaluated as above to avoid refining fixed errors from the previous stage, and residues predicted to be in error are selected. Finally, an additional 200-step Monte Carlo trajectory is run with the $Z_{error}^{(i)}$ computing solely from $Z_{rama}^{(i)}$ to ensure the favorable Ramachandran geometry in models.

## Pre-proline Ramachandran potential

Following early experiments, a new term was added to Rosetta that enforces a distinct pre-proline Ramachandran potential, replacing the original 20 different potentials:

$$E_{rama} = P(\varphi_i, \psi_i | AA_i)$$

With 40 different potentials conditioned on the sequence identity of the C-terminal adjacent residue:

$$E_{rama} = P(\varphi_i, \psi_i | AA_i, is\_pro_{i+1})$$

This potential was trained using the Richardson 8000 set of high-resolution crystal structures (*Chen et al., 2010*), and smoothed using adaptive kernel density estimates, as with the original Ramachandran potential (*Ting et al., 2010*). They are included in the released Rosetta with the energy term *rama_prepro* (using the same weight as the Rosetta term *rama*). *Figure 1—figure supplement 4* illustrates the resulting potentials. For all experiments in this manuscript, this term replaced the default Ramachandran score term in Rosetta.

## Local relax

Following our four cycles of refinement, we run a modified version of *Relax*, which we call *LocalRelax*. Modifications were made following the observation that – when applied to very large complexes (800+ residues) – we observed many instances in which sidechains were not properly optimized into density, even though the density was very clear. *Figure 5—figure supplement 1* shows several such cases.

In *Local Relax*, small overlapping regions of ~20–100 residues (discontinuous in sequence space) are selected for optimization repeatedly, until the entire protein has been optimized at least once. The approach is based upon the idea of neighbor residues, where residue neighbors are defined as all residues with a Cβ-Cβ distance less than 8Å. We first find the residue $r_i$ with the most residue neighbors. Then we optimize the neighbors of $r_i$, and the neighbors-of-neighbors of $r_i$: the neighbors are allowed to optimize both sidechain and backbone conformation, while the neighbors-of-

neighbors may only optimize sidechain conformation. This optimization is performed via Monte Carlo sampling of sidechain rotamers, followed by Cartesian minimization of all movable atoms. Following this, all neighbors of $r_i$ (as well as $r_i$) are marked as visited, and the process repeats, selecting a new $r_i$ as the unmarked residue with the most neighbors. This process continues until all residues are marked. In total, four cycles of this procedure are carried out, increasing the weight on van der Waals repulsion in each cycle. Finally, following coordinate refinement with *LocalRelax*, we fit atomic B-factors following the scheme described in our previous paper (*DiMaio et al., 2015*).

## Sidechain rescaling

We compute a scale factor associated with each sidechain that describes how much contribution to the density score is made by each sidechain. The values were computed using the 3.3-Å reconstruction of the 20S proteasome (*Li et al., 2013*) and the 3.2-Å reconstruction of β-galactosidase (*Bartesaghi et al., 2014*). Models were refined into the density and real-space atomic B-factors were fit for each atom. We then converted the atomic B-factors to scale factors using the following transformation:

$$scale_{AA} \approx \frac{1}{B^{3/2}}$$

Scales were normalized such that the scale for all backbone atoms was equal to 1. To prevent overfitting, each sidechain was grouped into one of three classes, and all sidechains within a given group were given the average scale factor of the group. Finally, while maintaining the ratio of these three groups with respect to one another, we scaled the relative contribution of backbone versus sidechain density, and selected the best values on the basis of free FSC following refinement. The final values range from 0.66 to 0.78, and are tabulated in *Table 3*.

**Table 3.** Sidechain scaling factors used in automated Rosetta structure refinement.

| Sidechain | Raw data | Factor used |
|---|---|---|
| ARG | 0.84 | 0.66 |
| LYS | 0.84 | 0.66 |
| GLU | 0.85 | 0.66 |
| MET | 0.87 | 0.66 |
| ASP | 0.88 | 0.66 |
| CYS | 0.87 | 0.71 |
| GLN | 0.89 | 0.71 |
| HIS | 0.91 | 0.71 |
| ASN | 0.91 | 0.71 |
| THR | 0.94 | 0.71 |
| SER | 0.95 | 0.71 |
| TYR | 0.95 | 0.78 |
| TRP | 0.96 | 0.78 |
| ALA | 0.97 | 0.78 |
| PHE | 0.98 | 0.78 |
| PRO | 0.98 | 0.78 |
| ILE | 0.99 | 0.78 |
| LEU | 0.99 | 0.78 |
| VAL | 1.00 | 0.78 |

## Voxel-size refinement

To optimize the voxel size of a map used to refine the model, we fix the model coordinates, and compute the model density. We then refine the voxel size $v = [v_x, v_y, v_z]$ and the origin $o = [o_x, o_y, o_z]$ of the map density – fixing these parameters in the model density – to maximize the real-space correlation coefficient between the two:

$$CC(v, o) = \frac{\sum \rho_o(\bar{x}) \cdot \rho_c(\hat{I}_{v,o}(\bar{x})) - \sum \rho_o(\bar{x}) \cdot \sum \rho_c(\hat{I}_{v,o}(\bar{x}))}{\left(\sigma_o^2(\bar{x}) \sigma_c^2 + (\hat{I}_{v,o}(\bar{x}))\right)^{1/2}}$$

$$\hat{I}_{v,o}(x, y, z) = (o_x + x/a_x, o_y + y/a_y, o_z + z/a_z)$$

Here, $\rho_0$ refers to the experimental map and $\rho_c$ to the map derived from the model, while $\sigma_o$ and $\sigma_c$ refer to the standard deviations over the corresponding density maps. Sums are taken over the entire map. Off-grid density values are computed using cubic splines to interpolate the calculated density map. This function is optimized with respect to the voxel-size parameters using l-BFGS minimization; analytic derivatives are computed for CC with respect to $v$ and $o$, and the same cubic splines are used to calculate derivatives with respect to the calculated map. Voxel size may be refined isotropically or anisotropically (using either four or six total parameters); all experiments in this manuscript treated this refinement isotropically (that is, all three axes are scaled together).

In this report, although we carried out voxel-size refinement for each of the three targets, we found only minimal changes of voxel size in all the cases. For fair comparison, we report all of the model-to-map metrics using maps with the deposited voxel sizes.

## Refinement against the full reconstruction and model selection

The previously described protocol was run to generate 5000 independent trajectories. From these 5000 models, a set of 10 representative models is chosen, following the protocol outlined in *Figure 1*. We want our optimized models to be optimal simultaneously in terms of: (a) independent map agreement, (b) physically realistic geometry, and (c) agreement to the full reconstruction. The last of these three is necessary because the full reconstruction often features details that are not present in the independent half maps.

Independent-map FSCs were computed against the validation map – subject to the same sharpening scheme as the training map – using the *ComputeFSC* mover in Rosetta. The integrated FSC between 10 Å and the reported resolution (3.4 Å in all cases) of the map was used to assess agreement with the independent map. The script computes FSC after masking the map with a mask computed from the model and filtered to 12Å with the command line:

*density_tools.exe -in:file:s model.pdb -mapfile validation_map.mrc -mask_radius 12 -nresbins 50 -lowres 10 -hires 3.4 -verbose*

In the case of the mitochondrial ribsosome, each segmented domain map was evaluated separately. Of the 1000 generated models, the top 50 by independent map agreement are selected.

Next, we want to identify the models from this subset that are the most physically realistic. To do this, all 50 models are rescored with MolProbity (*Chen et al., 2010*), and the top 10 are selected. While computing similar features to the Rosetta energy, its slightly different implementation makes it a somewhat orthogonal measure for structure evaluation.

Finally, we want to use features from the full reconstruction to further improve the model, particularly bulky sidechains that may not be visible in the half-map reconstructions. However, when refining against the full reconstruction, we need to be careful not to overfit to the full reconstruction, as we no longer have an independent map with which to evaluate overfitting. We use two ideas to avoid overfitting in this case. First, we do not perform any fragment-based rebuilding with the full map, and instead we perform just two cycles of *LocalRelax* and B-factor refinement with the full map. Second, we use half maps to determine the optimal fit-to-density weight when refining against the full map. The weight is selected using the following relation, where the weight is chosen to maximize the following:

$$U = FSC_{free} - 0.004 \cdot E_i$$

Here, $E_i$ is the per-residue energy, and is included as additional regularization to avoid overfitting. The value of 0.004 was chosen to normalize the two based on the relative dynamic ranges of both terms.

The top 10 models from the previous selection are subject to refinement against the full map. The final model is then taken as the model with best integrated-FSC against the full reconstruction. Local deviation over all ten models is used to estimate model uncertainty. The per-residue structural variance of ensemble models is calculated using *Theseus* with the default command line (*Theobald and Wuttke, 2008*).

## Assembly of the mitochondrial ribosome

In the case of the mitochondrial ribosome, we refine separate models for each protein subunit. A final assembly step combines the full model. In this final assembly step, all subunits, plus the deposited nucleic acid chains are combined in a single model, and are subject to 2 cycles of *LocalRelax* against the full reconstruction.

## EMRinger score calculation

For each of the five models following model selection, EMringer was run using the command:

*phenix.emringer MODEL.pdb MAP.ccp4*

To calculate per-chain EMRinger scores, pdb files were first segmented by chain ID and then EMringer scores were calculated against the segmented pdb files.

EMRinger scores can be compared absolutely between structures, although model size and local resolution variation are sources of noise for the EMRinger score. Scores below one are indicators of suboptimal model to map agreement for structures better than 4-Å resolution, while a score around zero indicates no improvement beyond randomness.

## Phenix real-space refinement

Starting from the deposited model for each of the three targets, real-space refinement was carried out using the Phenix package (v. 2450) with a default setting using the command:

*phenix.real_space_refine MODEL.pdb MAP.mrc resolution=3.4*

For the case of Frh, ligand files were appended to the above command with cif files generated using the command:

*phenix.elbow ligands.pdb*

## Availability

All methods described are available as part of Rosetta Software Suite, using weekly releases after week 35, 2016. The Rosetta XML files and flags for running all the refinements discussed in this manuscript are included as *Supplementary files 1–3*. The scripts and the tutorial used for running the method described here is available at the website of the corresponding author (https://faculty.washington.edu/dimaio/files/density_tutorial_sept15_2.pdf).

## Acknowledgements

The authors thank Drs Alan Brown, Alexy Amunts and Venki Ramakrishnan for sharing the half maps of the mitoribosomal large subunit (EMD-2762) with us; the authors especially thank Dr Alan Brown for providing helpful comments on the Rosetta refined mitoribosome, in which the suggestions led to the new development of better optimizing sidechains in very large protein complexes. The authors thank Drs Metteo Allegretti and Janet Vonck for sharing the half maps of Frh (EMD-2513) with us. We thank Dr Erhu Cao for commenting on the refined TRPV1 model initially. We thank Dr Vikram Mulligan for help on using the '-auto_setup_metals' module that he developed to facilitate ligand setup in modeling Frh using Rosetta. We also thank the members of the RosettaCommons for the continuted developement of the Rosetta Software Suite. RY-RW is grateful to Dr David Baker for the support and for suggesting avenues of research. This work was supported by the US National Institutes of Health under award numbers R01GM098672 (YC), P50GM082250 (YC), 1P01GM111126 (YC), and GM110580 (JSF); the US National Science Foundation under award number STC-1231306 (JSF).

## Additional information

### Competing interests
YS: Co-founder of Cyrus Biotechnology, Inc., which will develop and market graphic-interface software for using Rosetta. The other authors declare that no competing interests exist.

### Funding

| Funder | Grant reference number | Author |
|---|---|---|
| National Institutes of Health | R01GM098672 | Yifan Cheng |
| National Institutes of Health | P50GM082250 | Yifan Cheng |
| National Institutes of Health | 1P01GM111126 | Yifan Cheng |
| National Institutes of Health | GM110580 | James S Fraser |
| National Science Foundation | STC-1231306 | James S Fraser |

The funders had no role in study design, data collection and interpretation, or the decision to submit the work for publication.

### Author contributions
RY-RW, Conception and design, Acquisition of data, Analysis and interpretation of data, Drafting or revising the article; YS, Conception and design; BAB, JSF, Performed the EMringer analysis, Analysis and interpretation of data; YC, Provided the TRPV1 half-map data set with various bfactor sharpening, and analyzed and interpreted the results on the TRPV1 refinement, Analysis and interpretation of data; FD, Conception and design, Analysis and interpretation of data, Drafting or revising the article

### Author ORCIDs
Ray Yu-Ruei Wang, http://orcid.org/0000-0001-5025-9596
Benjamin A Barad, http://orcid.org/0000-0002-1016-862X
James S Fraser, http://orcid.org/0000-0002-5080-2859
Frank DiMaio, http://orcid.org/0000-0002-7524-8938

## Additional files

### Supplementary files
• Supplementary file 1. Input files to carry out the TRPV1 structure refinement described in the manuscript. Structure refinement of TRPV1 using Rosetta involves two steps: (1) refinement of only the transmembrane regions, and (2) refinement of the full system, including the Ankyrin repeat domains. The package includes two folders, one for each of the two steps with the command lines and input files necessary to run TRPV1 structure refinement.

• Supplementary file 2. Input files to carry out the Frh structure refinement described in the manuscript. Structure refinement of Frh using Rosetta involves three steps: (1) refinement of the asymmetric unit without ligands present, (2) local refinement of the asymmetric unit with the ligands present, and (3) local refinement the full symmetric complex with ligands present. The package includes three folders, one for each of the three steps with the command lines and input files necessary to run Frh structure refinement.

• Supplementary file 3. Input files to carry out the Mitoribosome structure refinement described in the manuscript. Structure refinement of the case of Mitoribosome using Rosetta involves in two steps: (1) refinement of individual chains, and (2) local refinement the whole assembly. The package includes two folders, one for each of the two steps with the command lines and input files necessary to run Mitoribosome structure refinement.

## Major datasets

The following previously published datasets were used:

| Author(s) | Year | Dataset title | Dataset URL | Database, license, and accessibility information |
|---|---|---|---|---|
| Liao M, Cao E, Julius D, Cheng Y | 2013 | Structure of the capsaicin receptor, TRPV1, determined by single particle electron cryo-microscopy | https://www.ebi.ac.uk/pdbe/entry/emdb/EMD-5778 | Publicly available at EBI Protein Data Bank in Europe (accession no: EMD-5778) |
| Allegretti M, Mills DJ, McMullan G, Kuehlbrandt W, Vonck J | 2014 | Electron cryo-microscopy of F420-reducing [NiFe] hydrogenase Frh | http://www.ebi.ac.uk/pdbe/entry/emdb/EMD-2513/ | Publicly available at EBI Protein Data Bank in Europe (accession no: EMD-2513) |
| Brown A, Amunts A, Bai XC, Sugimoto Y, Edwards PC, Murshudov G, Scheres SHW, Ramakrishnan V | 2014 | Electron cryo-microscopy of human mitochondrial large ribosomal subunit | http://www.ebi.ac.uk/pdbe/entry/emdb/EMD-2762 | Publicly available at EBI Protein Data Bank in Europe (accession no: EMD-2762) |

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
