## [Decision Letter]

Thank you for submitting your article "Automated structure refinement of macromolecular assemblies from cryo-EM maps using Rosetta" for consideration by *eLife*. Your article has been favorably evaluated by John Kuriyan (Senior Editor) and three reviewers, one of whom is a member of our Board of Reviewing Editors and another is Sjors HW Scheres (Reviewer #2).

The reviewers have discussed the reviews with one another and the Reviewing Editor has drafted this decision to help you prepare a revised submission.

Summary:

This manuscript concerns atomic model refinement against maps obtained from single particle electron microscopy data at near atomic resolution (for EM structures determined in the ~ 4.5 to 3 Å range). This is an important topic in modern structural biology, as the recent surge in resolution of cryo-EM structures calls for improvements in the methods to position and refine atoms in these maps.

Here the authors use the Rosetta package which takes into account prior knowledge about protein structure. Although Rosetta has been previously applied to EM map refinement, the authors describe a number of improvements of the method. The authors implemented (1) procedures to detect regions where the fit of manually built models produced strain in the local geometry as a consequence of misfitting the map or as a consequence of inaccuracy of the magnification parameter. Other improvements include (2) voxel size refinement, which may resolve the magnification inaccuracy of cryoEM maps; (3) side chain down-weighting during refinement, which may be proper for cryoEM maps since the molecules are not packed and therefore side chains less well defined compared to backbone; (4) refinement against the full map at the final stage.

Although none of these approaches are individually entirely new, it is the combination of these methods that may become a useful tool for the structural biologist working with EM maps. However, the reviewers have identified a number of issues that need to be addressed before a final decision can be made.

Essential revisions:

1) The improved refinement methods should be compared to standard refinement approaches, since otherwise it is difficult to judge the improvement of the presented over existing methods. They should for example compare with standard phenix.refine real-space simulated-annealing refinement. The amount of computational time for both approaches should also be provided. Finally, the comparison should systematically assess the quality of the model fit throughout the model, not just by subjectively showing particular regions.

2) There is clearly a need to find a better estimate of the voxel size. It is tempting to optimize the voxel in the way described here and it has been done in the same way before as published by other groups (these references to previous work should be provided). However, if the model has errors (which it always has), the best modelmap correlation is not always obtained for the correct voxel size. One of the reviewers performed a simple test using a PDB structure (PDB ID 4AKE) and simulated a density with voxel size 1.0 Å. Errors were introduced to the model by adding Gaussian noise to the atomic coordinates, here with a σ of 3 Å. 50 of such randomized models were generated. The modelmap correlation was then plotted for different voxel sizes, showing mean and standard deviation for these 50 models. It turns out that the error of the voxel size optimization is of the same order of magnitude as the typical error of the estimate from the microscope (~1%). Thus, the proposed voxel size estimation method may not produce the true voxel size. This issue needs to be addressed in the manuscript.

3) Figure 4 is supposed to show the mismatch of the deposited model and density. One of the reviewers downloaded the deposited model (PDB 4CI0) and EM map (EMD-2513) and could not reproduce this figure. The deposited model actually fits well into the deposited map. Images are attached to this decision letter of a similar view with the same contour level given in the figure caption (0.065). The authors need to clarify this issue.

4) The authors may wish to further elaborate on two limitations that are briefly mentioned in the final paragraph of the Discussion:

A) How to handle maps that have dramatic differences in local resolution. For example, in the TrpV1 channel case, the authors had to refine part of the model in two steps against two parts of the map (which is only mentioned in the Methods).

B) How to determine the sharpening B-factor objectively before real-space refinement. The authors appear to have determined the sharpening B-factor subjectively (as mentioned in the first paragraph of the Methods section). But what criterion was used? This issue may become more difficult when the map has heterogeneous local resolutions.

5) How does the method handle missing residues in loops or disordered region that are very difficult to place manually? This would be a very useful tool.

6) Figure 3, right panel. How are these different models of the ensemble selected? By using some map correlation criterion to get the "best" fitted models?

7) The original and optimized voxel sizes for the F420 example is given in the manuscript as 1.320 and 1.326 Å. The changes in voxel sizes should also be given for the other examples.

[Editors' note: further revisions were requested prior to acceptance, as described below.]

Thank you for resubmitting your work entitled "Automated structure refinement of macromolecular assemblies from cryo-EM maps using Rosetta" for further consideration at *eLife*. Your article has been favorably evaluated by John Kiriyan (Senior Editor) and three reviewers, one of whom is a member of our Board of Reviewing Editors.

The manuscript has been improved but there is a remaining issue that needs to be addressed before acceptance, as outlined in the following.

The reviewers found the explanation on the issue previously raised about Figure 5 not satisfactory. The shift shown in Figure 5 is not a result of voxel size error, but is likely due to the fact that the authors of the F420 structure (PDBID 4ci0) did not carefully generate the biological assembly structure. Maybe there was a slight mismatch in the symmetry operators used to generate the entire assembly. If one fits each subunit rigidly into the density (e.g. in Chimera) then all subunits fit very well (also with the original voxel size of 1.320). Figure 5 and the "deposited" items in panels B and C should therefore be removed. The effect of changing the voxel size from 1.320 to 1.326 Ang is much smaller than the current figure suggests.

---

## [Author Response]

*Essential revisions:*

*1) The improved refinement methods should be compared to standard refinement approaches, since otherwise it is difficult to judge the improvement of the presented over existing methods. They should for example compare with standard phenix.refine real-space simulated-annealing refinement. The amount of computational time for both approaches should also be provided. Finally, the comparison should systematically assess the quality of the model fit throughout the model, not just by subjectively showing particular regions.*

This is a very good point. We have included a comparison (in terms of model geometry, agreement-to-data, and runtime) to phenix real_space_refine. We also added a paragraph to compare the results between Rosetta and the Phenix refinement. To systematically assess model quality, we look at per-residue fit-to-data for all cases and report the number of residues which show significant better density-fit for both methods, as well as aggregate statistics such as RSCC, iFSC, and EMringer score (Table 1).

*2) There is clearly a need to find a better estimate of the voxel size. It is tempting to optimize the voxel in the way described here and it has been done in the same way before as published by other groups (these references to previous work should be provided). However, if the model has errors (which it always has), the best model-map correlation is not always obtained for the correct voxel size. One of the reviewers performed a simple test using a PDB structure (PDB ID 4AKE) and simulated a density with voxel size 1.0 Å. Errors were introduced to the model by adding Gaussian noise to the atomic coordinates, here with a σ of 3 Å. 50 of such randomized models were generated. The model-map correlation was then plotted for different voxel sizes, showing mean and standard deviation for these 50 models. It turns out that the error of the voxel size optimization is of the same order of magnitude as the typical error of the estimate from the microscope (~1%). Thus, the proposed voxel size estimation method may not produce the true voxel size. This issue needs to be addressed in the manuscript.*

For the references, unfortunately, we were unable to find out the previous work the reviewers pointed out. If advised, we will add them in future edits. However, we make two comments in regards to voxel size refinement. First, previous studies using EM maps as molecular replacement targets showed that voxel spacing may be off by as much as several percent (Jackson et al., Nature Protocols, 2015). More importantly, we feel our iterative approach – cycling between refinement and voxel spacing optimization – captures some of the agreement between forcefield and voxel spacing. We performed a related experiment, which we feel more accurately captures model errors. Using the same target as above (4ake) we ran short Rosetta MD trajectories followed by minimization; these cycles were repeated until models fell ~3Å RMS from the native structure. The result was 50 models each 2.9-3.1Å from native with relatively good energy.

Our voxel refinement procedure was then applied to each of these models, first optimizing voxel spacing/origin to maximize agreement with the model, and then running our iterative procedure. Five models were generated for each of the 50 starting models, and the one with highest agreement to the data is indicated. It yields the distributions shown below (true voxel size = 1.0Å).

Following refinement, the standard deviations on voxel size are reduced from 0.011 to 0.004; the latter numbers are increased by a few outlier structures for which the iterative refinement got stuck. This has been added as Figure 2 in the manuscript. And we added a paragraph describing the experiment.

*3) Figure 4 is supposed to show the mismatch of the deposited model and density. One of the reviewers downloaded the deposited model (PDB 4CI0) and EM map (EMD-2513) and could not reproduce this figure. The deposited model actually fits well into the deposited map. The authors need to clarify this issue.*

This figure was generated by downloading the PDB of the full biological unit and comparing it to the map. Even though the deposited asymmetric unit matches nicely with the map, the full biological unit poorly matches with the map. This is probably a minor issue, where the origin of the biological unit is inconsistent with the origin of the map; however, we were pleased to see that our voxel refinement identified this issue.

This error was first identified when we were trying to evaluate the biological unit with full symmetric partners present using EMRinger, where it was observed that the EMRinger score varied significantly between subunits.

4) The authors may wish to further elaborate on two limitations that are briefly mentioned in the final paragraph of the Discussion:

*A) How to handle maps that have dramatic differences in local resolution. For example, in the TrpV1 channel case, the authors had to refine part of the model in two steps against two parts of the map (which is only mentioned in the Methods).*

*B) How to determine the sharpening B-factor objectively before real-space refinement. The authors appear to have determined the sharpening B-factor subjectively (as mentioned in the first paragraph of the Methods section). But what criterion was used? This issue may become more difficult when the map has heterogeneous local resolutions.*

We agree that this is an important question in model refinement. However, a full solution for these problems is beyond the scope of this manuscript. We have made a few comments in the main text on the splitting of TrpV1 into two segments.

The refinement seems to be somewhat insensitive to differing levels of B factor sharpening. For example, the results (integrated FSC) for the 20S proteasome over different values of B-factor sharpening are plotted below (and added to the manuscript as Figure 2); in a range of sharpening values from –40 to –130, refinement performs equally well, even though subjectively, the maps appear different.

Consequently, we expect the refinement to be somewhat tolerant of small changes to local resolution; TrpV1 is an extreme case, where the repeat protein – particularly beyond the first two repeats – is of very low local resolution and is not tolerated by refinement. This is readily apparent upon running the full refinement and looking at model convergence.

*5) How does the method handle missing residues in loops or disordered region that are very difficult to place manually? This would be a very useful tool.*

This manuscript is only concerned with fixing local errors in hand-traced models; we have previously developed tools (RosettaCM) for rebuilding missing regions in density (e.g., Song et al. 2014 Structure.).

*6) Figure 3, right panel. How are these different models of the ensemble selected? By using some map correlation criterion to get the "best" fitted models?*

The models are selected using the criterion specified in the manuscript (best 50 by free FSC, best of these by MolProbity), as well as in Figure 1, Step 2. This has been clarified in the figure text.

*7) The original and optimized voxel sizes for the F420 example is given in the manuscript as 1.320 and 1.326 Å. The changes in voxel sizes should also be given for the other examples.*

This has been added in the Methods section. In all other cases, voxel spacing changed by <0.5% so the models that used were those refined without voxel spacing refinement.

[Editors' note: further revisions were requested prior to acceptance, as described below.]

*The manuscript has been improved but there is a remaining issue that needs to be addressed before acceptance, as outlined in the following.*

*The reviewers found the explanation on the issue previously raised about Figure 5 not satisfactory. The shift shown in Figure 5 is not a result of voxel size error, but is likely due to the fact that the authors of the F420 structure (PDBID 4ci0) did not carefully generate the biological assembly structure. Maybe there was a slight mismatch in the symmetry operators used to generate the entire assembly. If one fits each subunit rigidly into the density (e.g. in Chimera) then all subunits fit very well (also with the original voxel size of 1.320). Figure 5 and the "deposited" items in panels B and C should therefore be removed. The effect of changing the voxel size from 1.320 to 1.326 Ang is much smaller than the current figure suggests.*

We have followed this suggestion. Following the advice of the reviewers, we docked individual chains into density and re-refined from this starting point. The newly refined voxel size – while increasing slightly – ends up quite close to that of the deposited density map. Therefore, we chose to refine the structure with the deposited voxel size (1.320). We have removed the Figure 5 and panels as suggested by the reviewers, and highlight instead the geometry improvement of the model following refinement. The Frh statistics in Table 1 and Table 2 were updated, in response to the new experiments from the new starting point. Finally, a new paragraph was added in the Discussion section, addressing the drop of fit-to-density (iFSC) in all three benchmark cases.